# Few-Shot Joint Multimodal Entity-Relation Extraction via Knowledge-Enhanced Cross-modal Prompt Model

## ABSTRACT

Joint Multimodal Entity-Relation Extraction (JMERE) is a challenging task that aims to extract entities and their relations from text-image pairs in social media posts. Existing methods for JMERE require large amounts of labeled data. However, gathering and annotating fine-grained multimodal data for JMERE poses significant challenges. Initially, we construct diverse and comprehensive multimodal few-shot datasets fitted to the original data distribution. To address the insufficient information in the few-shot setting, we introduce the **K**nowledge-**E**nhanced **C**ross-modal **P**rompt **M**odel (KECPM) for JMERE. This method can effectively address the problem of insufficient information in few-shot setting by guiding a large language model to generate supplementary background knowledge. Our proposed method comprises two stages: (1) a knowledge ingestion stage that dynamically formulates prompts based on semantic similarity guide ChatGPT generating relevant knowledge and employs self-reflection to refine the knowledge; (2) a knowledge-enhanced language model stage that merges the auxiliary knowledge with the original input and utilizes a transformer-based model to align with JMERE's required output format. We extensively evaluate our approach on a few-shot dataset derived from the JMERE dataset, demonstrating its superiority over strong baselines in terms of both micro and macro $F_1$ scores. Additionally, we present qualitative analyses and case studies to elucidate the effectiveness of our model.

## CCS CONCEPTS

• **Computing methodologies** → **Information extraction**; Computer vision tasks; • **Information systems** → *Users and interactive retrieval.*

## KEYWORDS

few-shot learning, multimodal information extraction, large language model

## 1 INTRODUCTION

Recent focus has intensified on two pivotal subtasks for building a multimodal knowledge graph: Multimodal Named Entity Recognition (MNER)[16, 32, 37, 41, 46] and Multimodal Relation Extraction (MRE)[43–45]. These tasks aim to leverage user-generated content

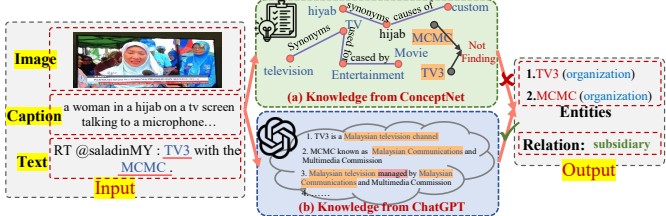

**Figure 1: Illustration of two different ways of acquiring auxiliary knowledge.**

from social media, primarily composed of images and text, to generate structured knowledge. However, previous works treated these tasks independently and restricted the exploration of their interconnections [32, 37, 43–46]. To address this problem, Yuan et al. [40] introduced a Joint Multimodal Entity-Relation Extraction (JMERE) task, aiming to enhance performance by integrating bidirectional interactions between the two subtasks. JMERE involves simultaneously extracting entities and their potential relations. However, previous JMERE work usually extracted features separately using pre-trained models (e.g., BERT for textes and ResNet for images), and employed multi-modal fusion layers for interaction, which typically require substantial training data [38]. Nevertheless, extensive multimodal data collection and annotation demand significant time and labor investments [47]. Furthermore, in real-world applications, access to labeled data is often constrained, and diverse domains necessitate specific datasets, presenting a challenge for data collection efforts. Thus, we first explore the JMERE task in few-shot setting, aiming to advance the practical applications of multi-modal information extraction. To our knowledge, we are the first to focus on JMERE in a multimodal few-shot scenario (FS-JMERE).

However, in few-shot setting, aligning different modalities and conducting information extraction between modalities pose heightened challenges compared to full datasets setting[38]. For instance, as illustrated in Fig. 1, discerning the relationship between **TV3** and **MCMC** based solely on the concise text "RT @saladinMY: TV3 with the MCMC." proves challenging, despite supplementary images hinting that both entities may belong to an organization such as television stations. An intuitive approach involves utilizing a large number of external Twitter data for pre-training, which can enhance the model's capacity for information extraction and relevant information storage [38, 39]. However, this approach requires extensive pre-train datasets and corresponding training times. Furthermore, some unimodality approaches aim to extract relevant information from sources such as knowledge graphs (e.g., Concept-Net [25]), to acquire background knowledge on entities, aiming to enhance the model's capability to extract entities or their relationships [10, 26]. However, these retrieval-based methods may encounter difficulties in locating relevant information and retrieving an excess of irrelevant information from knowledge graphs [21]. As shown in Fig. 1 section (a), **TV3** and **MCMC** are not present in

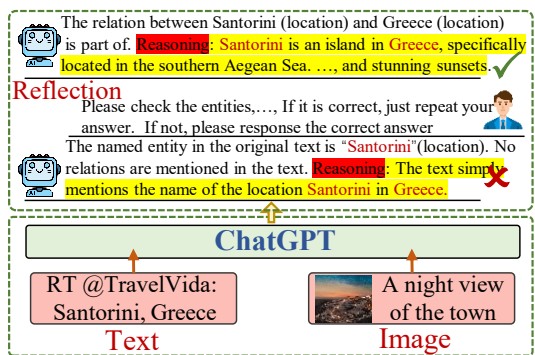

**Figure 2: An example of self-reflection within ChatGPT to correct its response.**

ConceptNet, and their absence introduces some irrelevant concepts knowledge, such as **movie** and **custom**. This information is not beneficial for extracting the entities of **TV3** and **MCMC** and their relationship.

Inspired by [21] and [11] using Large Language Models (LLMs) as an external knowledge repository for specified tasks, we aim to obtain relevant background knowledge from LLMs, such as *Malaysian television* managed by *Malaysian Communications and Multimedia Commission*, as depicted in Fig. 1 section (b). Such background knowledge can effectively address the challenge of insufficient information in few-shot setting without the need for additional pre-training. However, acquiring high-quality knowledge from LLMs relies on selecting appropriate prompt examples [12, 36]. Although one simplistic approach involves employing fixed contextual prompts, this method may generate prompts with very low relevance to given samples and obtain low-relevance responses from LLMs[36]. Yang et al. [36] indicated that all example selection strategies exhibit markedly inferior performance in comparison to the oracle strategy, which leverages the similarity of ground-truth answers. Furthermore, we have observed that responses from ChatGPT are not always definitive, potentially leading to inaccurate knowledge. As illustrated in Fig. 2, ChatGPT did not explicitly identify "Greece (location)" with the first response. However, it will correct itself and provide correct knowledge, after self-reflection [22]. Thus, it is crucial to guide LLMs in generating the necessary knowledge or addressing new tasks effectively by constructing more suitable contextual examples and responses [21, 30].

To address above challenges, we propose the **K**nowledge-**E**nhanced **C**ross-modal **P**rompt **M**odel (KECPM) for Few-shot Joint Multimodal Entity-Relation Extraction (FS-JMERE), as shown in Fig. 3. Initially, to create a few-shot training and development dataset, we sampled data according to distinct relationship types' data distribution. To address the insufficient information in FS-JMERE, we propose a method to acquire relevant knowledge from ChatGPT, which employs a dynamic prompt creation method and a knowledge selector to guide ChatGPT in generating more relevant auxiliary background knowledge. Formally, we manually annotate a limited set of human samples as candidate prompts, which comprises two primary components. The first part entails identifying the named entities and their relationships within the sentences. The second part involves offering comprehensive justifications by considering both

the image and text content, along with relevant knowledge. Subsequently, KECPM employs a multimodal similarity-aware module to select human samples and integrates them into prompt templates fitted for the JMERE task, thereby introducing relevant background knowledge. To further improve the relevance and usefulness of knowledge generated by ChatGPT, we enable multiple rounds of self-reflection [22], allowing it to rectify errors. Subsequently, we employ a knowledge selector to choose more accurate and relevant knowledge. This approach leverages ChatGPT's few-shot learning capability and its capacity for self-reflection within in-context learning. The auxiliary knowledge from ChatGPT's heuristic method is integrated with text and image captions, enhancing input for downstream models without requiring additional modality alignment for the FS-JMERE task. The main contributions of this study are summarized as follows:

- We focus on joint multimodal entity-relation extraction (JMERE) in the few-shot setting. To our knowledge, we are the first to focus on it and build the few-shot dataset by taking into account the distribution of relation categories.
- We propose a knowledge-enhanced cross-modal prompt model to address the few-shot scenario, utilizing dynamic prompts and knowledge reflection, to obtain background knowledge from LLMs for improved performance in FS-JMERE. Our approach enables the creation of contextually appropriate prompts, guiding LLMs to generate refined auxiliary knowledge efficiently. Additionally, we enhance relevance by iteratively selecting knowledge generated from LLMs through knowledge reflection.
- We perform comprehensive experiments on the constructed few-shot datasets, and the results show that our proposed model outperforms strong baselines in the JMERE in few-shot setting.

## 1.1 Related Work

## 1.2 Multimodal Information Extraction

*1.2.1 Multimodal Information Independent Extraction.* Previous works [15, 18, 42] used RNNs for text encoding and CNNs (e.g., VGG, Resnet) [7, 23] for unified image vector representation. For example, Lu et al. [15] proposed a gated network with CRFs regulating image-text interaction, and Zhang et al. [42] used bidirectional LSTM with attention to image-text alignment. However, such a method encoded the entire image as a vector and didn't distinguish different object types. Thus, recent studies use Mask-RNN techniques [6] to extract visual objects and align them with text by attention mechanism [16, 32, 37, 41, 46]. Wu et al. [32] proposed a dense co-attention network for MNER, integrating vital visual objects into text through cross-attention. To capture the relationship between entity information in images and text, recent methods explore establishing a graph structure between objects and entities [16, 41]. Zhang et al. [41] used a unified graph to represent sentence-image pairs and an extended GNN for multimodal semantic interactions. Lu et al. [16] created a multimodal interaction transformer with interaction position tags for unified representation and used a transformer for self-modal and cross-modal connections.

Compared to MNER, MRE is a relatively nascent research area, which aims to focus on extracting relationships between identified

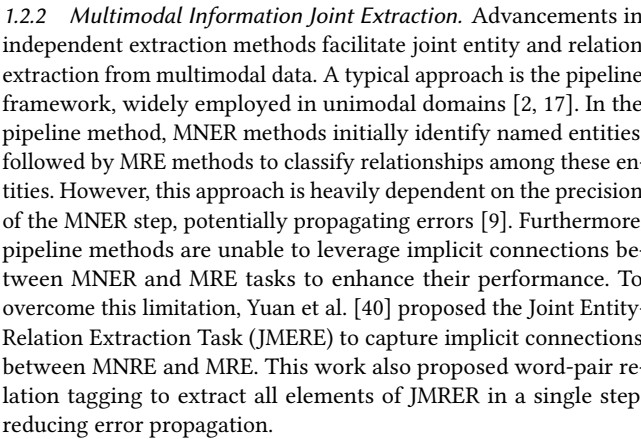

Figure 3: The KECPM architecture for few-shot JMERE comprises two stages: (a) The Knowledge Ingestion stage generates auxiliary knowledge from ChatGPT, pertinent to the provided text and image. This enhances the downstream model's contextual comprehension; (b) The Knowledge-enhanced LM combines the auxiliary knowledge with the original input, feeding it into a language model to address the few-shot JMERE task.

entities [43–45]. Zheng et al. [45] collected the first MNRE dataset from social media. Zheng et al. [44] proposed a graph alignment method that aligns structures from different modalities, like text syntax trees and object scene graphs, aiding in entity-object correspondence. To enhance entity relation extraction using visual data, Zhao et al. [43] introduced the two-stage visual fusion network (TSVFN), using multimodal fusion for improved relation extraction. However, these methods treated MNER and MRE tasks independently, thereby limiting the exploration of their interconnections.

*1.2.2 Multimodal Information Joint Extraction.* Advancements in independent extraction methods facilitate joint entity and relation extraction from multimodal data. A typical approach is the pipeline framework, widely employed in unimodal domains [2, 17]. In the pipeline method, MNER methods initially identify named entities, followed by MRE methods to classify relationships among these entities. However, this approach is heavily dependent on the precision of the MNER step, potentially propagating errors [9]. Furthermore, pipeline methods are unable to leverage implicit connections between MNER and MRE tasks to enhance their performance. To overcome this limitation, Yuan et al. [40] proposed the Joint Entity-Relation Extraction Task (JMERE) to capture implicit connections between MNRE and MRE. This work also proposed word-pair relation tagging to extract all elements of JMRER in a single step, reducing error propagation.

In this paper, we focus on the investigation of joint multimodal entity-relation extraction in few-shot setting. This choice aims to align with real-world applications that frequently offer only a restricted number of labeled data. We are the first to focus on JMERE in a multimodal few-shot scenario. Besides, We propose a knowledge-enhanced cross-modal prompt model to address the few-shot scenario, utilizing dynamic prompts and knowledge reflection, to obtain background knowledge from LLMs for improved performance FS-JMERE.

## 1.3 Few-shot unimodal Information Extraction

Established few-shot learning approaches have been applied in unimodal information extraction (e.g., Named Entity Recognition (NER)[3, 31, 35] and Relation Extraction (RE) [4, 5, 13]). The works in Few-shot Information Extraction (FS-IE) can be divided into two categories: the first category uses limited data for supervised training. Fritzler et al. [3] applied prototype networks [24] to address the few-shot NER (FS-NER) task. Inspired by nearest-neighbor inference in few-shot learning [31], Yang et al. [35] employed supervised feature extractors for FS-NER. Within the few-shot RE (FS-RE) task, Gao et al. [4] introduced attention mechanisms to augment prototype networks [24], and Han et al. [5] incorporated relationship description information as a means of distinguishing between complex and straightforward tasks. Liu et al. [13] incorporate relationship description with support sample representations, resulting in notably superior results. Nevertheless, this approach fails to address the issue of insufficient information in few-shot setting.

The second category focuses on pre-training or integrating external information to address the challenges of the few-shot setting. In the FS-NER task, Huang et al. [8] proposed a specialized noisy supervised pre-training approach, while Wang et al. [28] explored model distillation techniques to enhance few-shot NER performance. In the FS-RE task, Qu et al. [19] developed a comprehensive strategy involving the construction of a global relationship graph from Wikidata, which effectively facilitated the learning of posterior distributions of relation prototype vectors. Additionally, Yang et al. [33] aimed to improve model performance by introducing entity concepts.

However, constructing extensive unlabeled datasets and knowledge necessitates a significant human effort. Such an approach contradicts the initial goal of addressing the challenges posed by the few-shot setting. Besides, these methods have primarily been focused on the unimodal domain. Their direct application in the

multi-modal domain has the risk of loss of valuable image-related information potentially.

Compared with the above methods, We use LLMs as an implicit knowledge repository for the heuristic generation of additional textual knowledge. In contrast to the labor-intensive task of constructing extensive unlabeled datasets and knowledge repositories, this approach offers more inexpensive and precise knowledge. In addition, we introduce a supplementary knowledge generation method based on semantic similarity. This approach facilitates the dynamic generation of prompts, producing more contextually appropriate examples to guide ChatGPT in producing the requisite auxiliary background knowledge.

## 2 OUR PROPOSED MODEL

Following the JMERE task setting, the $i$-th input sample comprises a sentence denoted as $X_T^i$ and a corresponding image represented as $X_I^i$. The JMERE task aims to extract entities, including their respective types, while also determining the relationships that exist between these identified entities. The resulting output is structured as a collection of quintuples represented by $Y = \{(e_1, t_1, e_2, t_2, r)_c\}_{c=1}^C$, where $(e_1, t_1, e_2, t_2, r)_c$ pertains to the $c$-th quintuple comprising distinct textual entities $e1$ and $e2$. These entities are associated with their respective entity types $t_1$ and $t_2$, as well as the relationship denoted by $r$ between the two entities within the task context.

Fig. 3 presents the overall architecture of the proposed method, which comprises a conceptually straightforward two-stage framework. The Knowledge Ingestion stage is the crucial part of our proposed method. By constructing a cross-modal heuristic prompt, we use this prompt to guide ChatGPT to acquire initial predictions and background knowledge related to the JMERE sample. In the subsequent step, we aim to integrate additional supplementary knowledge into the language model, such as T5, to enhance the model's capacity to extract entities and relations between entities. We combine the aforementioned knowledge to augment the input text. This augmentation involves integrating type-based prompts and structured image descriptions as inputs to the language model, thereby further enhancing its capabilities. It is worth noting that we did not utilize visual object features (e.g., Mask R-CNN), because we found that incorporating these features did not lead to performance improvement. Detailed experimental setups and descriptions can be found in Section 3.7 Case Studies.

### 2.1 Stage-1: Knowledge Ingestion stage

This stage aims to generate auxiliary background knowledge from ChatGPT relevant to the given text and image, enhancing the downstream model's capacity to comprehend the context. This stage is divided into three steps: Human Prompts, Prompt Selection, and Knowledge Reflection. Subsequently, we describe each step in detail.

*2.1.1 Human Prompts.* To select suitable prompts for a given sample, we initially devised a human prompt set $P = \{p^1, p^2, \cdots p^M\}$ specifically designed for the JMERE task, where $M$ represents the number of predefined human prompt sets. Each human prompt $p^m$ includes not only the original text $p_T^m$ and its corresponding image description $p_I^m$, but also encompasses two distinct external knowledge components, as illustrated in Figure 3 (a): the first component

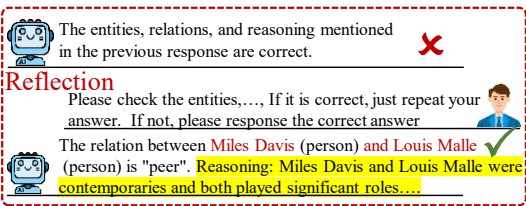

**Figure 4: An illustrative instance of self-reflection within ChatGPT reveals how this self-reflection process can occasionally yield unhelpful responses.**

$p_{k_1}^m$ delineates the entities and their corresponding relationships within the sentence, customized for the JMERE task. The second component $p_{k_2}^m$ provides background knowledge related to the text and images, with relevant background knowledge to support entity-relationship predictions. This knowledge and reasoning enhance the credibility of information extraction.

*2.1.2 Prompt Selection.* To ensure the relevance of prompts to the given sample, we developed a multimodal prompt-aware module to select appropriate examples from the human prompt set. As an essential component of multimodal knowledge extraction, the JMERE task necessitates the integration of textual and visual information. Therefore, we initially combine the original text $p_T^m$ and its corresponding image description $p_I^m$ as input into a language model $M_a$ (such as T5-large) to encode fused representations for each human prompt,

$$z^m = M_a([p_T^m; p_I^m]) \tag{1}$$

where $z^m \in \mathbb{R}^{d_h}$ represents the high-dimensional representation of the $m$-th human-prompt and $d_h$ denotes the output dimension of the language model. Similarly, for the $i$-th JMERE sample, we obtain its image description $\tilde{X}_I^i$ through image $X_I^i$ and its high-dimensional representation $q^i \in \mathbb{R}^{d_h}$ using the method in Eq. 1,

$$q^i = M_a([X_T^i; \tilde{X}_I^i]) \tag{2}$$

It is crucial to recognize that instances closer in high-dimensional space are likely to share more similar entity types and object information. Therefore, KECPM utilizes cosine similarity to compute the high-dimensional similarity between each JMERE sample and each predefined human prompt. Subsequently, KECPM selects the top-K most similar predefined human prompts as candidate prompts to be sent to ChatGPT for generating refined auxiliary knowledge.

$$\mathcal{I}_i^K = \underset{m \in \{1,2,\cdots M\}}{\arg \text{TopK}} \frac{(q^i)^T z^m}{\|q^i\|_2 \|z^m\|_2} \tag{3}$$

where $\mathcal{I}_i^K$ represents the top-K indices in human prompt set $P$ for $i$-th JMERE sample. Thus, the ultimate input for ChatGPT to query its knowledge comprises the in-context $C^i$, which is defined as follows:

$$C^i = \{(D, P^j, S^i) | j \in \mathcal{I}_i^K\} \tag{4}$$

| Statistics | | #S | #AL | Top 3 Relations | Last 3 Relations |
|---|---|---|---|---|---|
| FS-JMERE(*Seed 17*) | Train | 246 | 16.23 | peer(20%), member_of(19.36%), contain(13.33%) | religion(0.32%), parent(0.32%), neighbor(0.32%) |
| | Dev | 224 | 16.24 | member_of(19.06%), peer(18.73%), contain(12.71%) | alumi(0.33%), charges(0.33%), religion(0.33%) |
| FS-JMERE(*Seed 67*) | Train | 248 | 16.26 | peer(21.03%), member_of(18.47%), contain(13.27%) | neighbor(0.32%), alumni(0.32%), religion(0.32%) |
| | Dev | 230 | 16.31 | member_of(18.71%), peer(18.71%), contain(12.93%) | religion(0.34%), neighbor(0.34%), charges(0.34%) |
| FS-JMERE(*Seed 97*) | Train | 247 | 16.63 | peer(21.04%), member_of(18.45%), contain(13.27%) | neighbor(0.32%), alumni(0.32%), religion(0.32%) |
| | Dev | 231 | 15.94 | member_of(18.71%) peer(18.71%), contain(12.93%) | religion(0.34%), neighbor(0.34%), charges(0.34%) |
| JMERE(*full*) | Train | 3,618 | 16.31 | peer(21.32%), member_of(19.36%), contain(13.97%) | alumni(0.18%), religion(0.13%), race(0.13%) |
| | Dev | 496 | 16.57 | member_of(19.39%), peer(19.23%), contain(13.14%) | alumni(0.16%), race(0.16%), religion(0.16%) |
| | Test | 475 | 16.28 | peer (24.37%), member_of(17.18%), contain(15.47%) | charges(0.16%), siblings(0.16%), religion(0.16%) |

Table 1: The statistics of the FS-JMERE dataset. Here #S denotes the number of sentences. The #AL is the average length of the sentence. The elements in the Top 3 and the Last 3 Relations represent the relation type and their corresponding frequencies.

where $D$ represents a predefined head prompt, which describes the JMERE task in natural language according to the requirements, and $S^i$ is a text template for $i$-th JMERE sample, which is denoted as,

$$S^i = [\text{Context} :Text; \text{Image} :ImageCaption; \text{Answer} :]$$

2.1.3 *Knowledge Reflection.* The composed knowledge prompts $C^i$ send into ChatGPT. We encourage ChatGPT the autonomy to make its own judgments for each input sample by leaving the answer field blank, allowing ChatGPT to generate responses. However, we have observed that responses from ChatGPT are not always definitive, which could potentially lead to inaccuracies in the provided knowledge. Despite ChatGPT subtly hinting at its possession of relevant background knowledge, this may not guarantee correctness. As illustrated in Fig. 2, although ChatGPT did not explicitly identify "Greece (location)", it can be implicitly inferred from its reasoning that ChatGPT can recognize "Greece" as a city. Therefore, we encourage ChatGPT to reflect more deeply and acquire knowledge that is more pertinent and beneficial [22]. However, in practice, we have observed that while reflection can empower large language models to self-correct, it occasionally leads to meaningless responses, introducing considerable noise into the process. As shown in Fig. 4, for instance, we receive responses like *The entities, relations, and reasoning provided above are correct*, even inputting a description such as, "*If it is correct, just repeat your answer*".

To address these limitations, we propose a knowledge selector to choose meaningful auxiliary knowledge. After $N$ self-iterations in ChatGPT involving human interactions, we collect $N$ responses and process them using the same language model as Eq. (1). Consequently, the resulting representations of the ChatGPT responses (Auxiliary Knowledge) set are denoted as $R = \{r^1, r^2, \cdots, r^N\}$. We utilize cosine similarity to compute the high-dimensional similarity between each sample and every item of auxiliary knowledge, as shown in Eq. (3). Subsequently, we designate the most similar auxiliary knowledge as the final auxiliary knowledge, denoted as,

$$F_i = \underset{j \in \{1,2,\cdots N\}}{\arg \text{Top1}} \frac{(q^i)^T r^j}{\left\| q^i \right\|_2 \left\| r^j \right\|_2} \tag{5}$$

where $F_i$ is the final auxiliary knowledge for $i$-th input sample.

## 2.2 Stage-2:Knowledge-enhance LM

In this stage, we integrate auxiliary knowledge denoted as $F_i^1$ obtained from ChatGPT with the original input $q^i$. These combined inputs are fed into a transformer-based language model, such as T5, to generate both entities and their corresponding relationships. To formalize this process, we modify our training objectives to facilitate the generation of entity and relation information. To enhance the readability and interpretability of the generated outputs, we convert each element of the JMERE quintuples, two entities $e_1$ and $e_2$ with their corresponding types $t_1$ and $t_2$ and their relationship $c$, into a language sketch. For example, as shown in Fig. 3, the ground truth of the given sample is *"The relation between TV3 (organization) and Keadilan (organization) is subsidiary"*, where TV3 (organization) and Keadilan (organization) are two entities with their corresponding entity types, and the subsidiary describes their relation. It is important to note that when a sample contains multiple quintuples, we perform template filling for each quintuple, subsequently concatenating these templates using ";". Furthermore, the final formalized input adheres to the following format,

$$I = [\text{Typ} :TypePrompt; \text{Image} :ImageCaption; \\ \text{Knowledge} :knowledge; \text{Text} :Text]$$

$$H = \text{LanguageModel}(I) \tag{6}$$

where LM represents the Language Model and each element $h_i \in \mathbb{R}^{d_h}$ in $H$ represent each token representation of input $I$, where $d_h$ is the dimension of the language model. To optimize the model parameters, KECPM uses the language modeling loss as the loss function for the input sequence with ground-truth labels,

$$\mathcal{L} = -\sum_{t=1}^{|y|} \log p(y^t | H, y^{0:t-1}) \tag{7}$$

where $|y|$ is the length of output sequence $y$.

## 3 EXPERIMENTS

### 3.1 Few-shot Datasets

We conduct experiments on few-shot multimodal datasets built according to the distribution of relation categories from JMERE [40]. This dataset is currently the solely available resource for joint multimodal entity-relation extraction. In constructing datasets for

| Modality | Model | Micro | | | Macro | | |
|---|---|---|---|---|---|---|---|
| | | P | R | $F_1$ | P | R | $F_1$ |
| **Text** | Tplinker | 40.03±(2.51) | 22.91±(3.67) | 29.00±(2.86) | 19.40±(2.35) | 8.94±(1.56) | 12.21±(2.04) |
| | BiRTE ♣ | **50.69**±(3.19) | 26.95±(3.52) | 35.10±(3.43) | 23.89±(4.44) | 11.74±(1.48) | 15.70±(1.78) |
| | SpERT ♣ | 44.27±(2.13) | 32.78±(4.51) | 37.69±(3.02) | **25.19**±(2.35) | 16.05±(2.03) | 19.60±(1.78) |
| **Text-Image** | AGBAN+MEGA | 41.45±(2.87) | 27.02±(3.17) | 32.71±(3.02) | 24.71±(4.17) | 10.87±(1.34) | 15.10±(2.02) |
| | UMGF+MEGA | 42.31±(3.25) | 25.13±(1.97) | 31.53±(2.73) | 23.09±(4.01) | 9.98±(2.35) | 13.94±(3.57) |
| | EEGA ♠ | 47.27±(3.28) | 29.65±(2.65) | 36.44±(2.89) | 25.23±(3.35) | 13.31±(2.56) | 17.43±(2.89) |
| | KECPM | 44.06±(1.46) | **36.55**±(1.88) | **39.96**±(1.73) | 24.45±(2.12) | **18.85**±(2.49) | **21.29**±(2.34) |

Table 2: The average performance of various models for FS-JMERE is reported in terms of Micro and Macro metrics, where P, R, and $F_1$ represent precision, recall, and $F_1$-score, respectively. The ♣ denotes models specifically designed for few-shot setting. The ♠ signifies the state-of-the-art performance in the JMERE task with full training data, achieving a Micro $F_1$ score of 55.29.

FS-JMERE, a crucial consideration is to select a diverse set of samples that comprehensively cover various relation categories. We accomplish this by sampling data based on the distribution of relation categories within instances, aligning with the original distribution of relations. Detailed statistics of the dataset can be found in Table 1. For each dataset, we utilize a random sampling approach with three different seeds: {19, 67, 97}. This process results in the creation of three distinct sets of few-shot training and development datasets. It is important to note that each split is performed three times. Subsequently, we report the average performance and standard deviation derived from 9 training runs (3 × 3), providing a more robust evaluation of our model.

## 3.2 Baselines

Our baseline methods include joint entity-relation extraction in unimodality and multimodality. the implementation details for each method are described below.

### 3.2.1 Joint Entity-relation Extraction in Unimodality.

- **Tplinker** [29] is a one-stage method designed for the joint extraction of entities and overlapping relations. It accomplishes this by transforming the task into a Token Pair Linking problem, addressing specific questions regarding token positions and relations.
- **BiRTE** [20] represents a bidirectional extraction framework that leverages two complementary directions, incorporating a shared encoder component and an affine model for relation assignment. Moreover, it introduces a shared-aware learning mechanism aimed at alleviating convergence rate disparities.
- **SpERT** [1] employs a span-based approach to identify entities and relations within sentences, to improve the multi-word entity extraction.

### 3.2.2 Joint Entity-relation Extraction in Multimodality.

- **AGBAN+MEGA** is an enhanced iteration of AGBAN, originally introduced in [46]. It utilizes adversarial learning and a bilinear attention network to enhance the fusion and synchronization of entities with their corresponding visual objects, thus improving the performance of the MNER task.
- **UMGF+MEGA** is an enhanced version of UMGF described in [41], which integrates regional image features to represent objects and utilizes fine-grained semantic correspondences via transformer and visual backbones. This enhancement is

motivated by the recognition that informative object features are more crucial than the entire image for MNER task.
- **EEGA** is the state of the art to address the JMERE task. It leverages the interaction between the MNER and MRE tasks via word-pair relation tagging. Additionally, it introduces an edge-enhanced graph alignment network that effectively aligns text entities and objects by utilizing edge information across graphs.

## 3.3 Evaluation Metrics

To evaluate the performance of triplet extraction, we employ a set of evaluation metrics consistent with the JMERE setting [40]. These metrics include precision (P), recall (R), and micro $F_1$-score (Mic-$F_1$). Additionally, considering the imbalance in relationship labels within the dataset—where the top 3 labels account for 50% of the test set samples while the least 3 labels account for only 1%, we use the macro-$F_1$ score as an additional evaluation metric. This metric enables us to assess the model's effectiveness in learning from few-shot samples. The computation of macro-$F_1$ (Mac-$F_1$) is presented as follows:

$$\text{Mac–}F_1 = \frac{1}{N} \sum_{i=1}^{N} F_1(i) \tag{8}$$

where $F_1(i)$ represents the $F_1$ score for the $i$-th relation type, with $N$ representing the total number of relation types.

## 3.4 Implementation Details

During the knowledge ingestion stage, we utilize the OFA model [27], a multimodal pre-trained model used for converting images into corresponding text descriptions. Additionally, we employ T5-large as our feature extractor, facilitating the fusion of image captions and textual data. Furthermore, we utilize ChatGPT with the gpt-3.5-turbo version and set the sampling temperature to 0. In the knowledge-enhanced language model stage, we employ the AdamW optimizer [14] to minimize the loss function. To determine the optimal learning rate, we conduct a grid search within the range of $[2x10^{-4}, 2x10^{-6}]$, ultimately setting it to $5x10^{-5}$. Additionally, we incorporate a warm-up linear scheduler to regulate the learning rate. We set the maximum sentence input length to 500, while the mini-batch size is configured to be 6. Our model undergoes training for 30 epochs, and we select the model with the highest micro $F_1$-score on the development set for evaluation on the test

| Model | Micro | | | Macro | | |
|---|---|---|---|---|---|---|
| | P | R | $F_1$ | P | R | $F_1$ |
| T | 42.27±(1.78) | 35.09±(3.06) | 38.34±(2.22) | 21.56±(5.43) | 17.85±(4.06) | 19.53±(3.37) |
| T+I | 42.55±(3.20) | 35.74±(3.68) | 38.84±(3.42) | 21.41±(3.62) | 18.56±(3.48) | 19.88±(3.12) |
| T+I+$K_{Directly}$ | 42.65±(1.94) | 36.53±(2.45) | 39.35±(2.08) | 22.75±(2.02) | 18.26±(3.03) | 20.24±(2.41) |
| T+I+$K_{Reflection}$ | **44.10**±(1.58) | 36.13±(1.75) | 39.70±(1.71) | 24.23±(4.32) | 18.02±(3.29) | 20.62±(3.12) |
| *ours*(KECPM(T+I+$K_{Selected}$)) | 44.06±(1.46) | **36.55**±(1.88) | **39.96**±(1.73) | **24.45**±(2.12) | **18.85**±(2.49) | **21.29**±(2.34) |
| *ours*+OF | 42.78±(2.67) | 36.47±(3.03) | 39.37±(2.89) | 23.85±(4.76) | 18.28±(3.68) | 20.70±(3.27) |

**Table 3: Results of the ablation study for the Fs-JMERE task. The T, I, K, and OF represent original Text, Image Caption, Knowledge, and Visual Object Features.**

| Model | Micro | | | Macro | | |
|---|---|---|---|---|---|---|
| | P | R | $F_1$ | P | R | $F_1$ |
| ChatGPT$_{K=1}$♣ | 22.83 | 13.11 | 16.65 | 14.33 | 8.13 | 10.37 |
| ChatGPT$_{K=5}$♣ | 21.81 | 19.01 | 20.31 | 14.44 | 16.29 | 15.30 |
| ChatGPT$_{K=10}$♣ | 21.51 | 15.84 | 18.24 | 17.16 | 13.13 | 14.87 |
| KECPM$_{w/oPS(K=1)}$ | 43.58 | 35.68 | 39.24 | 24.03 | 17.82 | 20.46 |
| KECPM$_{w/oPS(K=5)}$ | 43.78 | 36.07 | 39.55 | **25.02** | 17.77 | 20.78 |
| KECPM$_{w/oPS(K=10)}$ | **45.02** | 35.07 | 39.43 | 23.02 | **19.24** | 20.96 |
| KECPM$_{K=1}$ | 43.82 | 35.87 | 39.45 | 24.35 | 17.89 | 20.63 |
| KECPM$_{K=5}$ | 44.06 | **36.55** | **39.96** | 24.45 | 18.85 | **21.29** |
| KECPM$_{K=10}$ | 44.21 | 35.92 | 39.64 | 24.35 | 17.89 | 21.17 |

**Table 4: Results of the number of in-context examples on auxiliary refined knowledge. The marker ♣ refers to ChatGPT with few-shot setting**

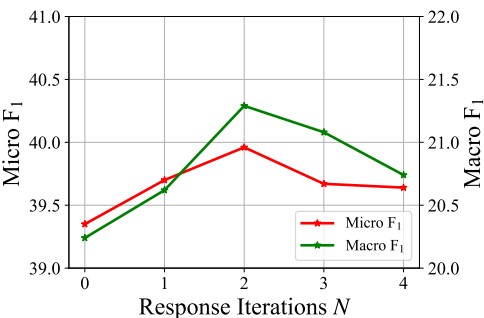

**Figure 5: The impact of iterations of self-reflection (N)**

set. All reported results are the average of three runs conducted with different random seeds.

## 3.5 Comparative Results

Table 2 presents the JMERE results on the few-shot setting, unveiling several crucial observations. Firstly, multimodal models, exemplified by AGBAN+MEGA, UMGF+MEGA, and EEGA, only achieved performance similar to unimodal methods like Tplinker and BiRTE, and even fell below the performance of SpERT, which was specifically designed for joint entity-relation extraction in the few-shot setting. This indicates that while the aforementioned multimodal models can obtain some visual features to benefit information extraction, they typically demand a significant number of data to align and fuse different modalities. This requirement contradicts few-shot setting, making it challenging to extract crucial information from visual features.

Furthermore, the SpERT model achieved notable performance in terms of Micro $F_1$ but under significant decline in Macro $F_1$ metrics, indicative of its struggle to effectively learn from long-tail data due to data imbalance. Moreover, our proposed model consistently outperforms all baseline models in both Micro $F_1$ and Macro $F_1$ metrics, showcasing its performance with improvements of 2.27 % and 1.69% in the respective metrics. Several factors contribute to the outstanding performance of our model: firstly, the utilization of an image caption-based approach enables the effective extraction of image information without the need for additional data training; secondly, the incorporation of refined auxiliary knowledge from ChatGPT equips the model with essential and relevant information, thereby enhancing overall performance and mitigating the impact of long-tail data; lastly, the employment of self-reflection

and knowledge selection further enhances the accuracy and relevance of knowledge information.

## 3.6 Ablation Experiments

Ablation studies were conducted to assess the effectiveness of each prompt in the proposed model: the original text, image caption, and three different methods for obtaining knowledge (Directly, Reflection, and Selected). The results of the ablation experiments are presented in Table 3. In comparison to using only the original text (**T**), the inclusion of Image Captions (**T+I**) notably improved the results, yielding a 0.5% increase in Micro $F_1$ and a 0.35% increase in macro $F_1$. This demonstrates that image information can offer insights into entity types. Moreover, the auxiliary knowledge of direct responses from ChatGPT (**T+I+K$_{Directly}$**) led to a substantial improvement in Macro $F_1$ (0.51%) and Micro $F_1$ (0.36%). This observation suggests that augmenting with auxiliary knowledge significantly impacts the performance, particularly in smaller sample sizes. Furthermore, through engaging ChatGPT in self-reflective (**T+I+K$_{Reflection}$**), we acquired higher-quality responses, resulting in an enhancement across both metrics. Nevertheless, when contrasted with our proposed model KECPM(**T+I+K$_{Selected}$**), there remains an opportunity for further improvement, particularly in the Macro $F_1$ (0.67%). This highlights the effectiveness of the proposed knowledge selector in filtering hallucinatory knowledge derived from multi-turn reflections.

Finally, we attempted to incorporate visual object features (OF) from Mask R-CNN into the input of the language model, similar to previous work [34], to include more fine-grained information. However, this integration did not result in performance improvement. We consider that aligning features across different modalities from distinct pre-training datasets (text from T5, objects from Mask R-CNN) demands a significant number of data, which is inappropriate in few-shot setting.

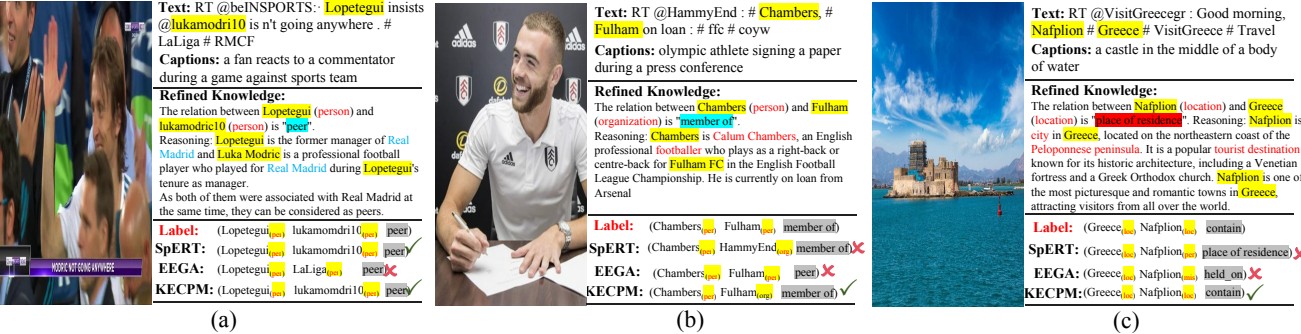

**Figure 6: Three case studies illustrating how auxiliary refined knowledge can aid in model predictions.**

## 3.7 Analysis of Prompt Selection and Knowledge Reflection Modules

This section explores the effectiveness of the Prompt Selection (PS) and knowledge reflection modules. *w/o* PS Means to remove the Prompt Selection module and replace it with random selection within the context. Additionally, we investigate the performance of ChatGPT in the JMERE task under the in-context learning setting.

The results, as shown in Table 4, Firstly, the ChatGPT with few-shot settings has a performance gap with our proposed model. We have observed that ChatGPTgenerates entity and relationship types not present in the FS-JMERE pre-defined, such as "building." This underscores the importance of Stage 2 in our proposed method, as it effectively aligns with the FS-JMERE task while leveraging background knowledge from ChatGPT. Moreover, the PS module can notably enhance the quality of auxiliary knowledge generated by ChatGPT and improve the model performance. Besides, an optimal number of human-prompt instances can further refine the quality of auxiliary knowledge. However, merely incorporating human prompts without consistent improvement in performance may not suffice. Supplying more than 5 prompts (K=10) to ChatGPT may decrease the quality of auxiliary knowledge. In practice, it has been observed that an excessive number of manual examples introduces noise into the generation process of ChatGPT, potentially disrupting its original reasoning process and leading to the replication of manual examples.

Finally, our proposed model integrates $N$ iterations of ChatGPT responses, enhancing knowledge accuracy through self-reflection. Consequently, we investigate the impact of varying iterations of self-reflection ($N$) on performance, as depicted in Fig. 5. When $N = 0$, which indicates the absence of self-reflection, the model exhibits the lowest performance, underscoring the efficacy of self-reflection in rectifying incomplete knowledge for improved performance. However, when $N$ exceeds 2, the model's performance gradually declines. Despite the inclusion of the proposed Knowledge Selector, which filters knowledge for relevance to the input text, multiple rounds of self-reflection also introduce more noise in responses.

## 3.8 Case Studies

Through three case studies depicted in Figure 6, we demonstrate how auxiliary refined knowledge enhances the predictive performance of the model. In the first case, only EEGA incorrectly identifies **LaLiga** as a *person*, attributed to the presence of only humans

in the image, leading to erroneous information fusion. Our model benefits directly from refined auxiliary knowledge, which accurately provides entity recognition and relationships. In the second case, the brevity of the text makes acquiring information about **Fulham** challenging, resulting in incorrect labeling by both SpERT and EEGA. However, our proposed model, with refined auxiliary knowledge included, effectively identifies the entity's type from such supplementary information.

The third case emphasizes the importance of integrating and fine-tuning auxiliary knowledge for the FS-JMERE task. While refined auxiliary knowledge correctly identifies entities **Greece** and **Nafplion** and their respective types, it incorrectly identifies the relationship between **Greece** and **Nafplion**. In the JMERE task, entities labeled as a *person* and an *organization* cannot have a *member of* relationship. By integrating this knowledge and fine-tuning the model, our method uncovers potential mappings between entity types and relationships, correcting errors and aligning with the FS-JMERE task requirements more effectively.

## 4 CONCLUSION

In this paper, we propose a Knowledge-Enhanced Cross-modal Prompt Model (KECPM) for Few-shot Joint Multimodal Entity-Relation Extraction (FS-JMERE), which can effectively handle JMERE in low-data scenarios by guiding a large language model to generate auxiliary background knowledge from text and image. To our knowledge, we are the first to focus on Multimodal Entity-Relation Extraction in the few-shot setting. Our model consists of two stages: firstly, a knowledge ingestion stage that dynamically creates prompts based on semantic similarity and uses self-reflection to refine the knowledge generated by ChatGPT; then, a knowledge-enhanced language model stage that integrates the auxiliary knowledge with the original input and employs a transformer-based model to perform JMERE. We have conducted extensive experiments on a few-shot dataset constructed from the JMERE dataset and demonstrated that our model outperforms strong baselines in terms of micro and macro $F_1$ scores. We have also provided qualitative analysis and case studies to illustrate the effectiveness of our model. For future work, we plan to explore more ways to generate and select high-quality auxiliary knowledge from ChatGPT, such as using reinforcement learning or adversarial learning. We also intend to apply our model to other multimodal information extraction tasks, such as multimodal sentiment analysis.

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
