# OpenReview forum: "Few-Shot Joint Multimodal Entity-Relation Extraction via Knowledge-Enhanced Cross-modal Prompt Model"
_acmmm.org/ACMMM/2024/Conference — MM2024 Poster_

### Official Review · Reviewer_2Uur · 2024-05-23

**Rating:** 4
**Confidence:** 3

**Summary:**

This paper concentrates on the task of Joint Multimodal Entity-Relation Extraction (JMERE), proposing a novel approach that incorporates large language models. The effectiveness of this method is validated through detailed experiments. While the task is innovative and the experiments are thorough, the architectural design lacks significant innovation. Additionally, there is a need for further discussion on efficiency and resource consumption to enhance the practical applicability of the proposed method.

**Strengths:**

1. The paper focuses on a current hot topic, integrating large language models for entity-relation extraction, which is highly relevant.
2. The experiments in the paper are comprehensive, including ablation studies and case studies that effectively demonstrate the method's efficacy.

**Limitations:**

1. While combining large models with traditional methods enhances accuracy, efficiency and resource consumption are equally critical for practical applications. This paper does not discuss the method's efficiency or resource utilization, which are essential factors to consider. It is recommended to include this analysis.

2. The architecture's utilization of multimodal information is somewhat limited, as is the comparison of cross-modal information. It would be worthwhile to explore the effectiveness of native large multimodal models on this task.

3. It is suggested that additional experiments be conducted to explore the improvements brought about by using different large model foundations.

4. The overall clarity and presentation of the manuscript could be enhanced. Additionally, some figures require improvement; for example, Figure 6 is distorted due to stretching, and the arrows in Figure 2 are confusing and do not clearly contribute to understanding the core contributions of the paper. Revising these figures is recommended.

**Suitability:**

3

---

### Official Review · Reviewer_TWZU · 2024-05-25

**Rating:** 2
**Confidence:** 3

**Summary:**

This paper build the few-shot dataset on joint multimodal entity-relation extraction(JMERE) task in the few-shot setting and propose a Knowledge-Enhanced Cross-modal Prompt Model (KECPM) which can effectively handle JMERE in low-data scenarios by guiding a LLM to generate auxiliary background knowledge from text and image. The results show that the model outperforms strong baselines in the JMERE in few-shot setting.

**Strengths:**

1.	Build the few-shot dataset by taking into account the distribution of relation categories.
2.	Propose a knowledge-enhanced cross-modal prompt model to address the few-shot scenario, utilizing dynamic prompts and knowledge reflection, to obtain background knowledge from LLMs for improved performance in FS-JMERE.

**Limitations:**

1.	The paper doesn't effectively utilize the content within the images, merely converting them into text using captioning tools. Moreover, the accuracy is reliant on the performance of the captioning tool. Ablation study also indicate that the results don't improve significantly with the addition of images.
2.	With the introduction of additional knowledge, the results don't seem to offer significant advantages compared to the baseline method.

**Suitability:**

3

---

### Official Review · Reviewer_DBET · 2024-05-25

**Rating:** 4
**Confidence:** 2

**Summary:**

The paper address the challenge of extracting entities and their relations from text-image pairs in social media posts with minimal labeled data. This task, known as Joint Multimodal Entity-Relation Extraction (JMERE), typically requires large datasets, which are difficult to gather and annotate. The authors propose a novel method called the Knowledge-Enhanced Cross-modal Prompt Model (KECPM) to effectively perform JMERE in a few-shot setting.

The KECPM approach consists of two main stages:

1. Knowledge Ingestion Stage: This stage dynamically generates prompts based on semantic similarity and uses ChatGPT to generate supplementary background knowledge. The knowledge is refined through self-reflection to improve accuracy.
2. Knowledge-Enhanced Language Model Stage: In this stage, the auxiliary knowledge is merged with the original input and fed into a transformerbased model to align with JMERE's output requirements.

The authors evaluated their model on a few-shot dataset derived from the JMERE dataset, demonstrating its superiority over existing baselines in terms of micro and macro F1 scores. The approach showed significant improvements in both precision and recall, highlighting the effectiveness of integrating background knowledge generated by large language models.

**Strengths:**

1. The introduction of a knowledge-enhanced crossmodal prompt model is a novel way to address the lack of data in few-shot learning scenarios. The dynamic prompt generation and knowledge refinement through self-reflection are particularly innovative.
2. The authors conducted extensive experiment on a few-shot dataset, showing that their model outperforms strong baselines in both micro and macro F1 scores.
3. The paper successfully integrates text and image information, addressing the complexity of multimodal data processing in social media contexts.

**Limitations:**

1. It seems that the model performance is highly dependent on the quality of the initial few-shot samples and the generated prompts. How to evaluate their performance while the author emphasizes the final result？
2. While the self-reflection mechanism is designed to refine knowledge, it can also introduce noise. This noise can lead to inaccuracies and affect the overall performance of the model, particularly if the reflection process does not yield high-quality knowledge consistently.
3. typo error: the authors use x instead of \times in line 690.

**Suitability:**

3

---

### Official Review · Reviewer_q6Fu · 2024-05-27

**Rating:** 6
**Confidence:** 3

**Summary:**

This paper introduces the Knowledge-Enhanced Cross-modal Prompt Model (KECPM) for JMERE. This method effectively addresses the problem of insufficient information in few-shot settings by guiding a large language model to generate supplementary background knowledge. The proposed method comprises two stages: (1) a knowledge ingestion stage that dynamically formulates prompts based on semantic similarity, guiding ChatGPT to generate relevant knowledge and employing self-reflection to refine this knowledge; and (2) a knowledge-enhanced language model stage that merges the auxiliary knowledge with the original input, utilizing a transformer-based model to align with JMERE’s required output format. The authors extensively evaluate the approach on a few-shot dataset derived from the JMERE dataset, demonstrating its superiority over strong baselines in terms of both micro and macro F1 scores. Additionally, the authors present qualitative analyses and case studies to elucidate the effectiveness of our model.

**Strengths:**

This paper we introduces a new Knowledge-Enhanced Cross-modal Prompt Model (KECPM) for JMERE.

Extensive experiments demonstrate that the proposed approach can obtain better performance.

This work presents qualitative analyses and case studies to elucidate the effectiveness of the model.

**Limitations:**

Some symbols are used without definitions, which makes certain parts difficult to follow.

The experimental analysis can be enhanced to provide more insights.

Missing references:

Rethinking Multimodal Entity and Relation Extraction from a Translation Point of View

Good Visual Guidance Makes A Better Extractor: Hierarchical Visual Prefix for Multimodal Entity and Relation Extraction

**Suitability:**

2

---

### Meta-Review · Area_Chair_YHLi · 2024-06-28

**Recommendation:** Accept (Poster)
**Confidence:** 5

**Metareview:**

The paper proposes a Knowledge-Enhanced Cross-modal Prompt Model for Joint Multimodal Entity-Relation Extraction in few-shot settings. The method addresses the challenge of insufficient data by guiding a large language model to generate supplementary background knowledge through dynamic prompts and self-reflection, which is then integrated with the original input using a transformer-based model. Extensive experiments demonstrate its superiority over strong baselines in terms of both micro and macro F1 scores. Reviewers praised the innovative approach, comprehensive experiments, and effective integration of text and image information. However, concerns were raised about the limited use of image content, reliance on the quality of initial samples and generated prompts, potential introduction of noise through self-reflection, and lack of discussion on efficiency and resource consumption. The authors provided satisfactory responses to some concerns during rebuttal. Despite these limitations, the overall quality and significance of the work justify acceptance.